# Stimuli-Sensitive Platinum-Based Anticancer Polymer Therapeutics: Synthesis and Evaluation In Vitro

**DOI:** 10.3390/pharmaceutics17111433

**Published:** 2025-11-05

**Authors:** Kateřina Běhalová, Martin Studenovský, Kevin Kotalík, Rafal Konefal, Marek Kovář, Tomáš Etrych

**Affiliations:** 1Institute of Microbiology of the Czech Academy of Sciences, v.v.i., Vídeňská 1083, 14220 Prague, Czech Republic; katerina.behalova@biomed.cas.cz (K.B.); makovar@biomed.cas.cz (M.K.); 2Institute of Macromolecular Chemistry of the Czech Academy of Sciences, v.v.i., Heyrovského sq. 2, 16206 Prague, Czech Republic; kotalik@imc.cas.cz (K.K.); rafal.konefal@amu.edu.pl (R.K.); etrych@imc.cas.cz (T.E.); 3NanoBioMedical Centre, Adam Mickiewicz University, Wszechnicy Piastowskiej 3, 61-614 Poznan, Poland

**Keywords:** cisplatin, polymer conjugates, HPMA copolymer, cancer chemotherapy

## Abstract

**Background/Objectives:** Here, we report the design, synthesis, and in vitro biological evaluation of a novel stimuli-sensitive nanotherapeutics based on cisplatin analog, *cis*-[PtCl_2_(NH_3_)(2-(3-oxobutyl)pyridine)] (Pt-OBP), covalently linked to a *N*-(2-hydroxypropyl)methacrylamide (HPMA) copolymer via a pH-sensitive hydrazone bond. **Methods:** Two polymer–drug conjugates, P-Pt-A and P-Pt-B, were synthesized, differing in spacer length between the polymer chain and hydrazone bond, which in turn modulates their drug release kinetics. **Results:** The spacer based on hydrazone bond demonstrated satisfactory stability under blood-mimicking conditions while enabling selective release of the active drug intracellularly or even in the mildly acidic tumor microenvironment. Pt-OBP exhibits comparable or even superior cytostatic and cytotoxic activity to carboplatin across a panel of murine and human cancer cell lines, with the highest potency observed in FaDu cells representing human head and neck squamous cell carcinoma. Mechanistically, Pt-OBP induced significant phosphorylation of γ-H2AX and activation of caspase-3, indicating its ability to cause DNA damage with subsequent apoptosis induction. P-Pt-A retained moderate biological activity, whereas the slower-releasing P-Pt-B exhibited reduced potency in vitro, consistent with its drug release profile. **Conclusions**: Notably, free Pt-OBP induced rapid apoptotic cell death, surpassing carboplatin at early time points, and the polymeric conjugates achieved comparable pro-apoptotic activity after extended incubation, suggesting effective intracellular release of the active drug.

## 1. Introduction

Conventional platinum agents such as cisplatin, carboplatin, and oxaliplatin are cornerstone chemotherapeutics used against a broad spectrum of malignancies. However, their clinical utility is hindered by severe side effects—including systemic myelotoxicity, or organ-specific toxicities like nephrotoxicity, ototoxicity, and neurotoxicity—as well as by acquired or intrinsic drug resistance [1,2]. To address these limitations, research over the past two decades has focused on the development of polymer–drug conjugates, aiming to enhance tumor targeting, improve pharmacokinetics, and minimize systemic toxicity [3,4]. The integration of platinum-based drugs into polymeric delivery systems constitutes a notable advancement in the field of oncological chemotherapy, offering the potential to improve therapeutic outcomes and reduce systemic toxicity.

Polymer therapeutics exploit the enhanced permeability and retention (EPR) effect, a phenomenon that allows macromolecules to preferentially accumulate in tumor tissues due to leaky vasculature and poor lymphatic drainage [5]. Pioneering work by Maeda and Matsumura [6] described the mechanism of EPR-based targeting, and subsequent efforts have sought to optimize polymer structure and drug release mechanisms. Commonly used polymer backbones include poly(*N*-(2-hydroxypropyl)methacrylamide); pHPMA), polyethylene glycol (PEG), and biodegradable polymers such as polylactide-co-glycolide (PLGA) and polyglutamic acid (PGA) [7,8]. These polymers are biocompatible and can be chemically modified to introduce active substances like drugs or imaging moieties, targeting ligands and stimuli-responsive linkers [9]. Among these, pHPMA-based copolymers are particularly well established as biocompatible drug carriers. They are highly hydrophilic, non-toxic in vivo (pHPMA was reported to be tolerated in rats at doses up to 30 g/kg [10]), non-immunogenic, and do not induce antibody formation [11]. In addition, HPMA copolymers exhibit stealth-like properties and do not significantly bind serum proteins [12], which further supports their safety and suitability for biomedical applications.

One of the most advanced systems, ProLindac (AP5346), is an pHPMA–oxaliplatin conjugate that demonstrated improved safety, enhanced water solubility, and preferential tumor accumulation in early-phase clinical trials [13,14]. Compared to free oxaliplatin, ProLindac showed reduced systemic toxicity and a more favorable pharmacokinetic profile, enabling dose escalation without exacerbating adverse effects. Another notable example includes PGA–cisplatin (PGA–CDDP), which progressed to Phase II clinical trials for the treatment of head and neck and ovarian cancers [15]. PGA–CDDP also provided enhanced antitumor efficacy while reducing nephrotoxicity and neurotoxicity commonly associated with cisplatin therapy. A micellar formulation of this conjugate, NC-6004, was later developed to further improve pharmacological performance. NC-6004 significantly extended the plasma half-life of cisplatin and demonstrated reduced kidney and nerve toxicity in preclinical models, while maintaining robust anticancer activity [16]. This system is currently being evaluated in clinical trials for multiple malignancies, both as monotherapy and in combination with other agents.

The nature of the linkage between the platinum complex and the polymer backbone is critical. Acid-labile, redox-sensitive, and enzymatically cleavable linkers have been explored to achieve controlled intracellular release of the active drug [17,18]. For instance, hydrazone bonds cleavable at acidic pH can exploit endosomal and lysosomal cleavage in cancer cells or even release the drug in mildly acidic tumor microenvironment. Similarly, disulfide linkers are reduced in the glutathione-rich cytosol of tumor cells, enabling site-specific release [19].

HPMA copolymer-based drug conjugates have attracted significant attention owing to their well-defined physicochemical properties, synthetic flexibility, and outstanding biocompatibility. Hydrazone linkages, in particular, have proven to be highly effective for pH-sensitive drug release, as they are relatively stable at physiological pH but readily cleaved in the acidic environment of endosomes and tumor tissues [17]. This property allows for selective drug release at the desired location, thereby enhancing the therapeutic index and minimizing systemic toxicity. Several studies have reported successful incorporation of hydrazone-linked drugs into HPMA backbones, yielding conjugates with favorable pharmacokinetics and tumor-targeting properties [11,18].

Polymer-based platinum therapeutics represent a promising strategy to overcome the limitations of conventional chemotherapy. Advances in polymer design, drug conjugation methods, and tumor-targeting strategies are poised to broaden their clinical applicability, potentially reshaping the treatment landscape for refractory cancers.

Here, we focused on the design, synthesis, and characterization of a novel stimuli-sensitive platinum-based anticancer agent covalently bound to a HPMA copolymer via a hydrazone bond. We introduced two novel HPMA copolymer conjugates bearing cis-[PtCl_2_(NH_3_)(2-(3-oxobutyl)pyridine)] (Pt-OBP) bound through pH-sensitive hydrazone bond with different drug release kinetics. We described the synthesis and physicochemical characterization of this conjugate as well as biological activity in several mouse and human carcinoma cell lines in vitro. Our new platinum-based drug released from the HPMA copolymer conjugate showed anticancer activity comparable to the carboplatin thus proving to be a promising anticancer drug. The conjugate with slower drug release kinetics showed a weaker cytostatic and cytotoxic activity in comparison to that with faster kinetics but holding a promise for higher anticancer activity in vivo.

## 2. Materials and Methods

### 2.1. Chemicals

1-Aminopropan-2-ol, 2-acetylpyridine, 2,2′-azobisisobutyronitrile (AIBN), 2-picoline, 2,4,6-trinitrobenzene sulfonic acid (TNBSA), 6-aminohexanoic acid, cisplatin, dimethyl sulfoxide (DMSO), formic acid, hydrazine hydrate, methacryloyl chloride, methyl 6-aminohexanoate hydrochloride, *N*,*N*-dimethylformamide (DMF), *N*-ethyldiisopropylamine (DIPEA), octyl pyrocatechin, silica gel, sodium hydride, *tert*-butyl carbazate, trifluoroacetic acid (TFA), and zinc powder were purchased from Sigma-Aldrich (St. Louis, MO, USA). Acetonitrile, acetic acid, ethyl acetate, methanol, and other common solvents and reagents were obtained from VWR Chemicals (Radnor, PA, USA) and used without further purification unless otherwise stated.

### 2.2. Analytical Methods

HPLC analyses were performed on a HPLC chromatograph (Shimadzu, Kyoto, Japan) using a reverse-phase column (Chromolyth Performance RP-18e 100 × 4.6 mm, Merck, Darmstadt, Germany) with UV detection. A mixture of water–acetonitrile in the presence of 0.1% TFA was used as the eluent at a gradient 0–100 vol % and a flow rate of 4 mL/min.

Elemental composition was determined using a Perkin Elmer Elemental Analyzer 2400 CHN (Perkin Elmer, Shelton, CT, USA).

Melting point temperatures were determined on a Kofler’s block (VEB Analytik Dresden, Dresden, Germany).

NMR spectra were measured on a Bruker Avance Neo 400 MHz NMR spectrometer (Bruker Daltonik, Bremen, Germany).

The molecular masses of low-molecular-weight compounds were determined using mass spectrometry on an LCQ Fleet mass analyzer with electrospray ionization (ESI-MS) (Thermo Fisher Scientific, Inc., Waltham, MA, USA).

Molecular weights of the polymers were determined by gel permeation chromatography (GPC) in a mixture of acetate buffer (pH 6.5, 0.3 mol/L) and methanol (20:80 *v*/*v*) on a TSK 3000 column (Polymer Laboratories Ltd., Shropshire, UK) at a flow rate of 0.5 mL/min using an HPLC System (Shimadzu, Kyoto, Japan) equipped with RI, UV and multi-angle light-scattering DAWN DSP-F detectors (Wyatt, Santa Barbara, CA, USA).

UV/VIS spectra were measured on a SPECORD 205 Spectrometer (Analytik Jena AG, Jena, Germany).

The platinum complex content in polymer conjugates was determined by HPLC as follows: 1 mg of the conjugate was dissolved in 1 mL of 1% aqueous HCl to ensure complete release of the platinum complex. After 30 min of incubation at room temperature, the resulting solution was directly analyzed by HPLC. The platinum complex concentration was calculated using an external calibration curve prepared from standard solutions of the free platinum complex (see Appendix A).

Platinum release from the polymer conjugates was studied in 1 mg/mL solutions incubated at 37 °C in three buffer systems: phosphate-buffered saline (PBS, pH 7.4), acetate–borate buffer (pH 6.5), and acetate buffer (pH 5.0). The amount of released platinum complex was quantified by HPLC and expressed as a percentage of the total drug content bound to the polymer at each time point.

### 2.3. Syntheses

#### 2.3.1. Ammonium Trichloroammineplatinate(II)

Synthesis of the starting platinum complex was performed according to the literature. [20].

#### 2.3.2. 2-(3-oxobutyl)pyridine

Synthesis of the pyridine-based ligand was performed according to the literature [21].

#### 2.3.3. cis-[PtCl_2_(NH_3_)(2-(3-oxobutyl)pyridine)] (Pt-OBP)

Ammonium trichloroammineplatinate(II) (90 mg, 0.28 mmol) was dissolved in 2 mL of 20% aqueous NaCl solution. Subsequently, 2-(3-oxobutyl)pyridine (60 mg, 0.4 mmol) was added, and the reaction mixture was heated at 50 °C for 48 h. Upon cooling to room temperature, a yellow precipitate formed, which was collected by filtration, washed successively with water and methanol, and dried under vacuum. Yield: 45 mg (37%). MS (ESI): a molecular ion peak corresponding to [M+H]^+^ was detected at *m*/*z* = 432. ^1^H NMR (400 MHz, DMSO-d6): *δ* = 9.13–9.08 (m, 1 H, Ar–-C*H*=N-), 8.00–7.94 (m, 1 H, Ar), 7.56–7.52 (m, 1 H, Ar), 7.42–7.37 (m, 1 H, Ar), 4.49–4.08 (br s, 2 H, -NH_2_), 3.68–3.62 (m, 2 H, -C*H*_2_-CH_2_-C(=O)-), 2.96–2.85 (m, 2 H, -CH_2_-C*H*_2_-C(=O)-), 2.18 (s, 3 H, C*H*_3_-C(=O)-) ppm.

#### 2.3.4. *N*-(2-hydroxypropyl)methacrylamide (HPMA)

The monomer was prepared as described in the literature [22].

#### 2.3.5. 6-Methacrylamidohexanohydrazide (MAHH)

The monomer was prepared as described in the literature [23].

#### 2.3.6. Poly(HPMA-co-MAHH)

HPMA (3.2 g, 22.4 mmol), MAHH (165 mg, 0.77 mmol), and AIBN (150 mg, 0.9 mmol) were dissolved in 20 mL of methanol. The solution was purged with dry argon for 20 min and subsequently heated in a sealed ampoule at 60 °C for 17 h. The resulting copolymer was precipitated into ethyl acetate, collected by filtration, reprecipitated from methanol into ethyl acetate, and dried under vacuum. Yield: 2.7 g (80%); Mw = 30,000 g mol^−1^, *Đ* = 2.2 (GPC); hydrazide group content (TNBSA assay [24]): 0.14 mmol g^−1^ (2 mol%).

#### 2.3.7. *N*-(Terc-Butoxycarbonyl)-*N*′-(Methacryloyl)hydrazine (MAH-Boc)

*Tert*-butyl carbazate (12.6 g; 0.096 mol; 1 equiv.) was dissolved in DCM, followed by addition of sodium carbonate (12.17 g; 0.12 mol; 1.2 equiv.) and small amount of polymerization inhibitor 1,1,3,3-(tetramethylbutyl)pyrocatechol. After cooling to 0 °C, methacryloyl chloride (9.35 mL; 0.096 mol; 1 equiv.) in 25 mL DCM was added dropwise. The reaction then proceeded at r.t. for 2 h. Sodium carbonate was filtered and the filtrate was washed with water (3 × 50 mL), the organic layer was dried, concentrated and the product was crystallized after addition of hexane. Yield: 13.4 g (70%). M. p. 118 °C. elemental analysis (calculated wt.%/found wt.%): C 53.98/53.94; H 8.05/8.06; N 13.99/14.06, ^1^H NMR (400 MHz, DMSO-d6): *δ* = 9.65 (s, 1 H, -*N*H-C(O)-C-), 8.71 (s, 1 H, -N*H*-C(O)-O-), 5.70 (s, 1 H, -C=C*H*_2_), 5.41 (s, 1 H, -C=C*H*_2_), 1.86–1.84 (m, 3 H, -C*H*_3_), 1.40 (s, 9 H, -O-C-(C*H*_3_)_3_) ppm. ^13^C NMR (101 MHz, CDCl3): *δ* = 167.8 (-NH-*C*(O)-C-), 155.9 (-NH-*C*(O)-O-), 137.9 (-*C*=CH_2_), 121.5 (-C=*C*H_2_), 82.0 (-O-*C*-(CH_3_)_3_), 28.3 (3 C, -O-C-(*C*H_3_)_3_), 18.5 (-*C*H_3_) ppm.

#### 2.3.8. Poly(HPMA-co-MAH)

This polymer precursor was synthetized in two steps. First, monomers HPMA and MAH-Boc (molar ratio HPMA:MAH-Boc = 94:6; concentration of monomers 16.0 wt. %) were polymerized by solution free radical polymerization using AIBN initiator (concentration 1.0 wt. %) as follows: HPMA (2000 mg, 0.014 mol) and MAH-Boc (179 mg, 0.89 mmol) were dissolved in methanol (11.3 mL), AIBN (136 mg, 0.83 mmol) was dissolved in DMA and added and the solution was inserted into an ampoule and bubbled with Ar (10 min). The polymerization was carried out at 60 °C for 17 h. The polymer was isolated by precipitation into a mixture of acetone and diethyl ether (300 mL, 2:1, *v*/*v*), filtered and dried under vacuum. Reprecipitation was performed from methanol. Yield: 1613 mg (74%). For deprotection of MAH groups, the following procedure [25] was used: the polymer was dissolved in 16.1 mL of deionized water (10% solution, *w*/*v*), transferred into an ampoule and bubbled with Ar (10 min). The ampoule was heated to 100 °C for 1 h. The final product was obtained by lyophilization. Yield: 1446 mg; M_w_ = 20,000 g mol^−1^; M_w_/M_n_ = 1.54 (GPC); MAH content: 5 mol% (^1^H NMR).

#### 2.3.9. Poly(HPMA-co-MAHH) Conjugate with Pt-OBP (P-Pt-A)

*Pt-OBP* (90 mg, 0.21 mmol) was dissolved in 1.5 mL of dimethylformamide and added to a solution of poly(HPMA-co-MAHH) (400 mg, 0.056 mmol of hydrazide groups) in 2 mL of acetic acid. The reaction mixture was stirred at room temperature for 30 min. The resulting polymer conjugate was precipitated into 10 mL of a diethyl ether/dichloromethane mixture (1:1, *v*/*v*). The crude product was collected by centrifugation, reprecipitated twice from a methanol solution into the same ether/dichloromethane mixture, and dried under vacuum. Yield: 430 mg; M_w_ = 57,000 g mol^−1^; M_w_/M_n_ = 1.3 (GPC); platinum complex content: 5.4 wt% (HPLC).

#### 2.3.10. Poly(HPMA-co-MAH) Conjugate with Pt-OBP (P-Pt-B)

*Pt-OBP* (70 mg, 0.16 mmol) was dissolved in 1.0 mL of dimethylformamide and added to a solution of poly(HPMA-co-MAH) (400 mg, 0.11 mmol of hydrazide groups) in 2 mL of acetic acid. The reaction mixture was stirred at room temperature for 150 min. The resulting polymer conjugate was precipitated into 10 mL of a diethyl ether/dichloromethane mixture (1:1, *v*/*v*). The crude product was collected by centrifugation, reprecipitated twice from a methanol solution into the same ether/dichloromethane mixture, and dried under vacuum. Yield: 450 mg; M_w_ = 89,000 g mol^−1^; M_w_/M_n_ = 1.95 (GPC); platinum complex content: 4.0 wt% (HPLC).

### 2.4. Cell Lines

FaDu (human hypopharyngeal carcinoma), LL2 (murine Lewis lung carcinoma) and 4T1 (murine mammary carcinoma) cell lines were purchased from American Type Culture Collection (ATCC, Manassas, VA, USA). SCC7 (murine squamous cell carcinoma) cell line was kindly provided by Dr. Deanne M.R. Lathers from the Medical University of South Carolina (Charleston, SC, USA). 4T1 and SCC7 cells were cultured in RPMI-1640 medium (Thermo Fisher Scientific, Waltham, MA, USA) supplemented with 10% heat-inactivated fetal bovine serum (FBS), 100 U/mL penicillin and 100 U/mL streptomycin, for 4T1 cell line 1mM sodium pyruvate, 4.5 g/L glucose and 10 mM HEPES solution were also added. LL2 cells were cultured in Dulbecco Modified Eagle Medium (DMEM; Thermo Fisher Scientific, Waltham, MA, USA) supplemented with 10% FBS, 100 U/mL penicillin and 100 U/mL streptomycin, 4.5 g/L glucose and 10 mM HEPES solution. For FaDu cell line, Eagle’s Minimal Essential Medium (EMEM, Media supply center of the Institute of Molecular Genetics, Prague, Czech Republic) containing 10% FBS, 100 U/mL penicillin and 100 U/mL streptomycin, 10 mM HEPES solution, 1 mM sodium pyruvate and 0.15% of sodium bicarbonate was prepared. Cells were cultured in the atmosphere of 5% CO_2_ at 37 °C in CO_2_ incubator (PHCbi, Tokyo, Japan). Cells were passaged three times a week.

### 2.5. [^3^H]-Thymidine Incorporation Assay

4.0 × 10^3^ (4T1, LL2, SCC7) or 8.0 × 10^3^ (FaDu) cells were seeded into flat-bottom 96-well plates (Thermo Fisher Scientific, Waltham, MA, USA). Various concentrations of tested compounds were added. Medium alone was added to control wells. After 72 h of incubation in CO_2_ incubator (5% CO_2_, 37 °C), [^3^H]-thymidine (PerkinElmer, Shelton, CT, USA) was added at a final dilution of 1:1500 for 6 h. Cells were harvested on a membrane (1450-421 Printed Filtermat, PerkinElmer, Shelton, CT, USA) and sealed in plastic bags (1450-432 Sample Bag, PerkinElmer, Shelton, CT, USA) using a cell harvester (Harvester 96, TOMTEC, Unterschleissheim, Germany). Scintillation detector (Microbeta2 2450 Microplate Counter, PerkinElmer, Shelton, CT, USA) was used for the measurement. In control cell samples, activity exceeding 3 × 10^4^ cpm was always measured. Results are shown as IC_50_ values (the concentration of the tested compound causing 50% inhibition of cell proliferation). At least three independent experiments were performed. Samples were measured in tetraplicates.

### 2.6. MTT Assay

4.0 × 10^3^ (4T1, LL2, SCC7) or 8.0 × 10^3^ (FaDu) cells were seeded into flat-bottom 96-well plates (Thermo Fisher Scientific, USA). Various concentrations of tested compounds were added. Medium alone was added to control wells. After 72 h of incubation in CO_2_ incubator (5% CO_2_, 37 °C), plates were centrifuged (4 °C, 357× *g*, 5 min) and the supernatant was discarded. 120 μL of 5 mg/mL (3-(4,5-Dimethylthiazol-2-yl)-2,5-Diphenyltetrazolium Bromide) (MTT) was added to wells for another 2 h. 200 μL of dimethyl sulfoxide (Sigma-Aldrich, Burlington, MA, USA) was added to each well for 15 min to dissolve formazan formed by metabolic processing of the tetrazolium salt by metabolically active cells. Absorbance was measured using microplate reader (Infinite 2000 PRO, TECAN, Männedorf, Switzerland) at 530 nm with reference at 690 nm. Results are shown as IC_50_ values. At least three independent experiments were performed. Samples were measured in tetraplicates.

### 2.7. Caspase-3 Assay

FaDu cells (5 × 10^5^/sample) were seeded into 10 cm Petri dishes in 9.5 mL of sterile culture media. Tested samples were added in 500 μL of PBS. PBS only was added to control samples. After the 48 h of incubation the cells were harvested, transferred into 50 mL tubes and centrifuged (4 °C, 357× *g*, 5 min). To determine the levels of activated Caspase-3, EnzChek™ Caspase-3 Activity Assay Kit (Thermo Fisher, Waltham, MA, USA) with Z-DEVD-AMC substrate was used. Briefly, the cells were washed with 10 mL of PBS and 1 × 10^6^ of cells from each sample were incubated with 230 μL of lysing buffer for 75 min on ice. The cells were centrifuged (4 °C, 5000× *g*, 5 min) and supernatants were transferred to clean tube. 50 μL of cell lysates were transferred into flat bottom 96-well plate (Thermo Fisher, Waltham, MA, USA) in triplicates, 50 μL of 2 × substrate working solution (20 μL 10 mM Z-DEVD-AMC substrate, 400 μL 5 × reaction buffer (supplied), 10 μL 1M DTT and 590 μL of dH_2_O for 1 mL) were added to each well. The plate was incubated for 30 min in the dark at room temperature. Fluorescent intensity was measured (excitation: 342 nm; emission 441 nm; gain manual 95; 25 flashes; bottom reading mode) using microplate reader (Infinite 2000 PRO, TECAN, Männedorf, Switzerland). Data are shown as a mean of at least 3 independent experiments.

### 2.8. Annexin-V Assay

FaDu cells (5.0 × 10^5^ for 48 h incubation or 2.5 × 10^5^ for 72 h incubation) were seeded into 6-well plates (Thermofisher, Waltham, MA, USA). Various concentrations of the tested compounds were added. After 48 or 72 h of incubation in the CO_2_ incubator (5% CO_2_ at 37 °C), cells were harvested, filtered through 30 μm strainer (Celltrics^TM^, Sysmex, Kobe, Japan) and centrifuged (4 °C, 357× *g*, 5 min). Pellets were resuspended in 100 μL of Annexin binding buffer (ABB; 10 mM HEPES, 140 mM NaCl, 2.5 mM CaCl_2_, pH 7.4) and transferred to 96-well conical bottom plate (MicroWell™ Plates, Thermofisher, Waltham, MA, USA). Cells were centrifuged (4 °C, 357× *g*, 5 min) and washed twice with 200 μL of ABB. Cells were stained with 20 μL of 1:50 solution of Annexin V conjugated with Dyomics 647 (EXBIO, Vestec, Czech Republic). After 30 min of incubation on ice in the dark, 100 μL of ABB was added to wells. Cell suspension was transferred to 96-well U-shaped bottom plates (Tissue culture test plate 96U, TPP, Trasadingen, Switzerland). Before the flow cytometry measurement, 20 μL of 50 mg/mL propidium iodide (PI) were added to each well. The measurement was performed using flow cytometer (LSR II, BD, San Jose, CA, USA), 5 × 10^4^ live cells were analyzed for each sample. Data analysis was carried out in the FlowJo^TM^ software (version 10.8.1). Data are shown as a mean of at least 3 independent experiments.

### 2.9. γ-H2AX

FaDu cells (2.5 × 10^5^) were seeded into 6-well plates (Thermofisher, Waltham, MA, USA). Various concentrations of the tested compounds were added. After 24 h of incubation in the CO_2_ incubator (5% CO_2_ at 37 °C), cells were harvested, filtered through 30 μm strainer (Celltrics^TM^, Sysmex, Kobe, Japan) and centrifuged (4 °C, 357× *g*, 5 min). Pellets were washed twice with 2 mL of PBS and 1 mL of ice-cold 70% ethanol was added for the fixation. After 1 h of fixation, cells were centrifuged (4 °C, 1000× *g*, 5 min) and washed twice with 500 μL of FACS buffer (PBS, 2% FBS, 2 mM EDTA). Cells were transferred to 96-well conical bottom plate (MicroWell™ Plates, Thermofisher, Waltham, MA, USA) and stained with 100 μL of 1:50 solution of anti-γH2AX antibody conjugated with the PE fluorochrome (clone N1-431, BD, San Jose, CA, USA). After 30 min of incubation on ice in the dark, 100 μL of FACS buffer was added to wells. Cell suspension was transferred 96-well U-shaped bottom plates (Tissue culture test plate 96U, TPP, Trasadingen, Switzerland). The measurement was performed using flow cytometer (LSR II, BD, San Jose, CA, USA); 5 × 10^4^ live cells were analyzed for each sample. Data analysis was carried out in the FlowJo^TM^ software (version 10.8.1). Data are shown as a mean of at least 3 independent experiments.

### 2.10. Statistical Analysis

Statistical analysis was carried out using unpaired two-tailed Student’s *t*-test. Significant differences are shown *(*“*ns*” represents no significant difference; ** p* ≤ 0.05, ** *p* ≤ 0.01, **** p* ≤ 0.001, **** *p* ≤ 0.0001).

## 3. Results and Discussion

### 3.1. System Design

In this work, we designed two stimuli-sensitive polymeric conjugates in which a low-molecular-weight platinum(II) complex of the cisplatin type, Pt-OBP, was covalently attached to an *N*-(2-hydroxypropyl)methacrylamide (HPMA) copolymer through a pH-sensitive hydrazone linker. The design of polymer therapeutics aims to ensure adequate stability in the bloodstream while enabling drug release in response to acidic microenvironments characteristic of tumors or intracellular compartments. Two different hydrazide-containing monomers (MAHH and MAH) were used to explore the effect of linker structure on conjugate behavior, yielding P–Pt–A and P–Pt–B, respectively. Pt-OBP (see Figure 1) was designed as a platinum complex (a) capable of reversible conjugation to a polymeric carrier, and (b) anticipated to exhibit favorable biological behavior, potentially analogous to that of picoplatin. Picoplatin, a sterically hindered platinum(II) complex bearing a 2-methylpyridine ligand, was developed to overcome key limitations of cisplatin, including resistance mediated by thiol-containing biomolecules and poor activity against certain solid tumors. It has demonstrated improved pharmacokinetics and reduced reactivity with intracellular thiols such as glutathione, thereby enhancing its cytotoxicity against resistant cancer cells [26]. Furthermore, picoplatin entered clinical trials for various malignancies, showing promising activity and a more favorable toxicity profile than first-generation platinum drugs [27].

### 3.2. Synthesis and Characterization of Platinum–Polymer Conjugates

Two polymer–drug conjugates were synthesized by coupling the cisplatin analog Pt-OBP to HPMA-based copolymers bearing hydrazide groups via an acid-cleavable hydrazone linkage. The conjugates, designated as P–Pt–A and P–Pt–B, differ in the chemical structure of the hydrazide-containing comonomer, leading to variations in the hydrolytic stability of the polymer–Pt–OBP conjugate. Synthetic routes of monomers, polymer precursors and related polymer–platinum conjugates are shown in Appendix A and Figure 2. The conjugation reactions were performed under mild acidic conditions (acetic acid, room temperature). The polymer–drug conjugates were obtained in almost quantitative yields (P–Pt–A: 430 mg/95%; P–Pt–B: 450 mg/97%) and were purified by repeated precipitation steps. The platinum complex Pt-OBP was successfully incorporated in sufficient amounts for further biological evaluation: 5.4 wt% for P–Pt–A and 4.0 wt% for P–Pt–B. GPC analysis revealed monomodal molecular weight distributions with weight-average molar masses (M_w_) of 57,000 g mol^−1^ for P–Pt–A and 89,000 g mol^−1^ for P–Pt–B. The corresponding dispersities (*Đ*) were 1.3 and 1.95, respectively. Chromatographic analyses confirmed the successful attachment of the platinum complex to both polymer carriers, resulting in stable conjugates suitable for subsequent evaluation of their pH-responsive release behavior.

### 3.3. pH-Responsive Platinum Release

The pH-dependent release profiles of Pt-OBP from P–Pt–A and P–Pt–B were evaluated in buffer solutions simulating the blood (pH 7.4), the extracellular tumor milieu (pH 6.5), and lysosomal compartment (pH 5.0). Both conjugates exhibited pH-sensitive behavior, releasing platinum significantly faster at acidic pH compared to pH 7.4. While P–Pt–B showed relatively slow release under physiological conditions (~25% after 24 h), P–Pt–A released almost 60% of its payload at pH 7.4 within 24 h, indicating limited systemic stability for this conjugate. Interestingly, the structural differences between the two copolymers had a pronounced effect on the rate of platinum release. P–Pt–A, which contains a longer and more hydrophobic linker, showed faster drug release across all tested pH values, including physiological pH, suggesting lower overall stability. In contrast, P–Pt–B, bearing a shorter and more hydrophilic linker derived from MAH, exhibited slower and more selective pH-dependent release, particularly showing delayed platinum release kinetics under neutral conditions. This difference may arise from decreased hydration and reduced steric flexibility around the hydrazone bond in P–Pt–B, which could stabilize the linker and slow acid-catalyzed cleavage. While P–Pt–A may be better suited for applications requiring rapid intracellular release, P–Pt–B demonstrates greater promise for systemic delivery due to its improved stability at blood pH (Figure 3 and Figure 4).

To gain a deeper mechanistic insight into the release behavior of P–Pt–A and P–Pt–B conjugates, cumulative release data (0–24 h) were fitted using two common kinetic models: the Higuchi model (Q = k_H × √t; Q—cumulative amount of drug released at time t, k_H—Higuchi release constant, t—time) and the first-order exponential model (M_t_ = M_∞_ (1 − e^−kt^); Mₜ—cumulative amount of drug released at time t, M_∞_—total amount of drug released at infinite time, k—first-order release rate constant, t—time). Each data set was analyzed based on the mean of triplicate measurements, and the quality of fit was assessed by the coefficient of determination (R^2^ = 1 − Σ(yᵢ − fᵢ)^2^/Σ(yᵢ − ȳ)^2^; yᵢ are the observed data points, fᵢ are the fitted values, and ȳ is the mean of observed values. A value of R^2^ = 1 indicates a perfect fit.) R^2^. The first-order model assumes a release rate proportional to the concentration of the unreleased drug. In contrast, the Higuchi model describes drug release from a homogeneous matrix by Fickian diffusion. Across all tested conditions (pH 5.0, 6.5, and 7.4), the first-order model provided an excellent fit with R^2^ values > 0.9 in all data sets. Conversely, the Higuchi model showed poor correlation with the experimental data (R^2^ < 0). This discrepancy suggests that drug release is not governed by surface diffusion, as assumed in the Higuchi model, but rather by a rate-limiting hydrolysis reaction, consistent with the chemical nature of the hydrazone linker. The release profile exhibits an exponential behavior characteristic of pseudo-first-order kinetics, which is expected for hydrolytic bond cleavage in excess water under mild conditions.

These findings highlight the importance of precisely balancing linker hydrophobicity, length, and steric accessibility to fine-tune the pharmacokinetic behavior of polymer–drug conjugates. Rational optimization of these parameters can facilitate the design of systems tailored for specific therapeutic applications, ranging from rapid intracellular drug release to extended systemic circulation. The synthetic flexibility of the HPMA platform, combined with the tunability of hydrazone-based linkers, offers a robust foundation for the rational design of advanced stimuli-responsive polymer therapeutics.

### 3.4. Pt-OBP Has Comparable Cytostatic and Cytotoxic Activity to Carbo-Pt

To study the anticancer efficacy of the novel Pt-OBP derivative in vitro, we tested its cytostatic and cytotoxic activities in 4T1 murine mammary carcinoma, LL2 murine non-small-cell lung carcinoma, SCC7 murine squamous carcinoma and FaDu human hypopharyngeal carcinoma. Cis-Pt and Carbo-Pt were used as reference compounds. Pt-OBP showed comparable potential to Carbo-Pt in inhibiting the cancer cell proliferation across most selected models. FaDu cell line was the most sensitive (Figure 5A,C) with an IC_50_ value ~1.8 μM for both Pt-OBP and Carbo-Pt. SCC7 and LL2 cells were the least sensitive, with IC_50_ values 9.9 μM and 9.1 μM, respectively. We further assessed the cytostatic activity of Pt-OBP polymer conjugates (P-Pt-A and P-Pt-B) to determine whether they retain the antiproliferative potential of the parent compound. Cytostatic activity of the conjugates corresponds to the drug release kinetics, as P-Pt-A showed approximately twofold higher IC_50_ values compared to Pt-OBP, while P-Pt-B exhibited significantly reduced cytostatic potency, with IC_50_ values 2–5 times higher than P-Pt-A depending on the cell line.

The cytotoxic activity of Pt-OBP was also comparable to that of Carbo-Pt in most of the tested cell lines (Figure 5B,D). The highest cytotoxic effect was observed in the FaDu cell line, with IC_50_ values of 8.2 μM for Pt-OBP and 17.4 μM for Carbo-Pt. The polymeric conjugates P-Pt-A and P-Pt-B showed slightly reduced activity, with P-Pt-A retaining moderate cytotoxic potential and P-Pt-B showing IC_50_ values up to twofold higher compared to P-Pt-A.

The cytotoxic activity of the tested platinum-based compounds was, in general, lower in comparison with their cytostatic activity. Therefore, we hypothesize that low concentrations of the compound are capable of effectively leading to cell cycle arrest and block the proliferation of the cells; however, 2–5 × higher concentrations are required to induce the cytotoxic effect and cause the cell death. A similar trend has also been reported in the literature. For instance, in the case of Cis-Pt, substantial differences between cytostatic and cytotoxic activity were observed in several cell lines, such as HeLa (IC_50_ = 2.64 μM vs. 19.60 μM) and HepG2 (IC_50_ = 3.11 μM vs. 9.55 μM) following 24 h of incubation [28].

### 3.5. Pt-OBP Leads to Activation of Caspase-3 and Causes DNA Double Strand Breaks

To further evaluate the mechanism of action of the novel Pt-OBP derivative, we assessed its ability to induce DNA damage by measuring levels of phosphorylated γ-H2AX, a marker of DNA double-strand breaks that recruits and localizes DNA repair proteins. Platinum-based drugs are known to generate replication stress and DNA breaks in cancer cells [29]. Pt-OBP induced γ-H2AX phosphorylation in FaDu cells at concentrations starting around 100 μM, at levels comparable to Carbo-Pt. This DNA-damaging activity was retained in both P–Pt–A and P–Pt–B conjugates (Figure 6A). We further examined the ability of Pt-OBP to activate the caspase-3. Caspase-3 is the executioner caspase in both the intrinsic and extrinsic apoptotic pathway and its activation irreversibly leads to apoptotic cell death [30]. Pt-OBP induced a significant increase in activated caspase-3 levels in FaDu cells at a concentration of 100 μM. In comparison, the reference compound Carbo-Pt triggered a similar effect only at a higher concentration (320 μM). The polymeric conjugates P–Pt–A and P–Pt–B also activated caspase-3, although P–Pt–B exhibited slightly reduced potency corresponding to the drug release kinetics. These results indicate that Pt-OBP is more effective than Carbo-Pt in triggering apoptosis, and this pro-apoptotic activity is preserved after conjugation to HPMA-based polymers (Figure 6B).

### 3.6. Pt-OBP Potently Triggers Apoptotic Cell Death

Since we have already determined the cytostatic and cytotoxic activity of Pt-OBP and its polymeric conjugates and detected the potential to activate caspase-3 and cause double strand breaks in DNA, we confirmed the ability to trigger apoptosis in FaDu cells using Annexin-V assay. Using Annexin-V and PI staining, we detected percentage of live, early apoptotic and late apoptotic cells. After 48 h of incubation, Pt-OBP was much more potent inducer of apoptosis than Carbo-Pt and while Pt-OBP led to apoptosis of 50% of cells at concentration 20–40 μM, Carbo-Pt had only minor effect even at concentration 160 μM (Figure 7A,B). P-Pt-A and P-Pt-B exhibited reduced potential in apoptosis induction reflecting their drug release kinetics. Increasing the incubation time up to 72 h led to enhanced efficacy of Carbo-Pt (50% of apoptotic cells at concentration 20 μM), but the efficacy was still comparable to Pt-OBP (50% of apoptotic cells at concentration 20 μM) (Figure 7C,D). Thus, it seems that Pt-OBP is a comparable apoptosis inducer at longer incubation times while it is more potent at shorter incubations, i.e., its pro-apoptotic effects are expressed faster than those of Carbo-Pt.

This observation is consistent with the known pharmacokinetics of platinum-based drugs, where Carbo-Pt requires longer incubation to reach peak activity due to slower hydrolysis into its active aqua complex compared to Cis-Pt [31]. The fact that Pt-OBP was markedly more effective than Carbo-Pt at 48 h, but showed comparable pro-apoptotic activity after 72 h, may suggest that the release of its active species occurs faster than in Carbo-Pt, but slower than in Cis-Pt. As a result, Pt-OBP may provide an advantage in inducing apoptosis in tumors especially at earlier time points post-administration.

The activity in apoptosis induction was similar for both P-Pt-A and P-Pt-B at 72 h, indicating that most of the active drug had likely been released from both polymeric conjugates at this incubation time.

Based on our findings, we expect that the novel Pt-OBP derivative shares the mechanism of action with both Cis-Pt and Carbo-Pt. Lower anticancer activity of Carbo-Pt in comparison with Cis-Pt was observed in various cancer cell lines, e.g., ovarian cancer cell lines [32] or kidney cancer cells A498 [33]. The reduced in vitro anticancer activity likely results from different affinity of Carbo-Pt and Cis-Pt to membrane transporter Ctr1, which leads to different levels of its intracellular accumulation and DNA damage [34]. In our study, we present the platinum-based derivative Pt-OBP, which—based on our mechanistic investigations—appears to mimic the mode of action of cisplatin and carboplatin, yet demonstrates comparable or even superior activity relative to carboplatin. The observed differences in anticancer efficacy may arise from mechanisms analogous to those responsible for the differential activity between cisplatin and carboplatin. The polymeric conjugates retained the in vitro anticancer activity in all assays and the results correspond to drug release kinetics. The polymeric conjugates, namely P-Pt-B, could prolong the circulation time in the bloodstream due to the prolonged drug release, improve pharmacokinetics, enhance accumulation in solid tumors and therefore improve the anticancer efficacy of Pt-OBP.

Platinum-based chemotherapeutics, such as cisplatin, carboplatin, and oxaliplatin, remain first-line agents for the treatment of solid tumors including lung, ovarian, head and neck, and colorectal cancers [35]. However, their clinical efficacy is often restricted by systemic toxicity, narrow therapeutic windows, and the development of drug resistance [36]. Conjugation of platinum derivatives to HPMA-based polymeric carriers provides a promising strategy to overcome these drawbacks by enhancing tumor accumulation, reducing off-target effects, and enabling controlled pH-responsive drug release. In this context, our findings indicate that the novel Pt-OBP derivative, particularly in its HPMA-conjugated form, may represent a suitable platform for the development of next-generation platinum therapeutics with improved pharmacological performance. While further in vivo evaluation and comprehensive preclinical safety and efficacy studies will be required to determine its translational viability, the present data support the relevance of this polymer–platinum approach for future therapeutic development.

## 4. Conclusions

In this study, we successfully designed, synthesized, and characterized two stimuli-sensitive HPMA copolymer–platinum conjugates incorporating the Pt(II) complex cis-[PtCl_2_(NH_3_)(2-(3-oxobutyl)pyridine)] via an acid-sensitive hydrazone linkage. The conjugation strategy yielded polymer–drug conjugates with defined molecular weight distributions, satisfactory drug loading, and consistent release profiles in mildly acidic conditions.

We showed that the novel platinum (II) derivative Pt-OBP exhibits potent anticancer activity in vitro, comparable to or exceeding that of the clinically used drug carboplatin. Pt-OBP effectively inhibited cell proliferation and induced cell death in several cancer cell lines, including murine 4T1, LL2, and SCC7, as well as the human head and neck cancer cell line FaDu. Notably, FaDu cells were highly sensitive to Pt-OBP, with cytostatic and cytotoxic IC_50_ values comparable to those of cisplatin and lower than those of carboplatin.

Importantly, the biological activity of Pt-OBP was preserved after conjugation to HPMA copolymers, with both P–Pt–A and P–Pt–B conjugates showing pH-responsive drug release but with P-Pt-B being more stable at blood pH. These findings align with the known advantages of polymer–drug conjugates, including controlled release, prolonged circulation, and the potential for tumor-selective accumulation via the enhanced permeability and retention (EPR) effect. Collectively, these results validate Pt-OBP as a potent anticancer agent with a mechanism of action similar to established platinum-based drugs and demonstrate that its polymer conjugates preserve therapeutic activity while offering tunable release kinetics.

## Figures and Tables

**Figure 1 pharmaceutics-17-01433-f001:**
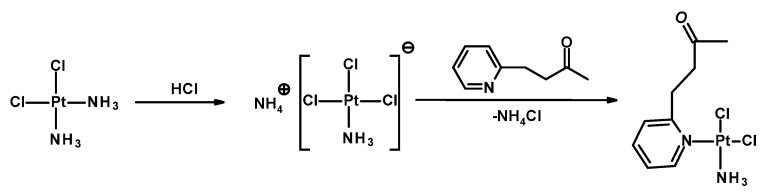
Scheme of synthesis of Pt-OBP.

**Figure 2 pharmaceutics-17-01433-f002:**
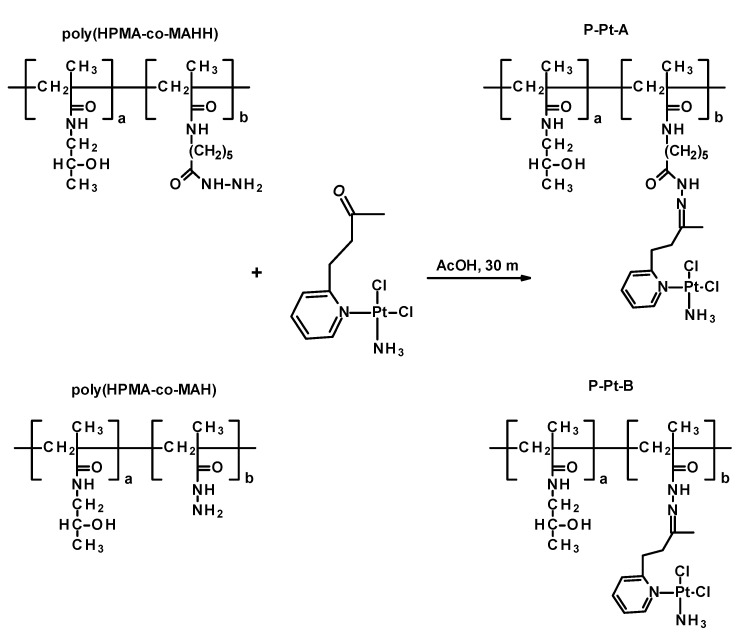
Scheme of synthesis of polymer conjugates P-Pt-A and P-Pt-B.

**Figure 3 pharmaceutics-17-01433-f003:**
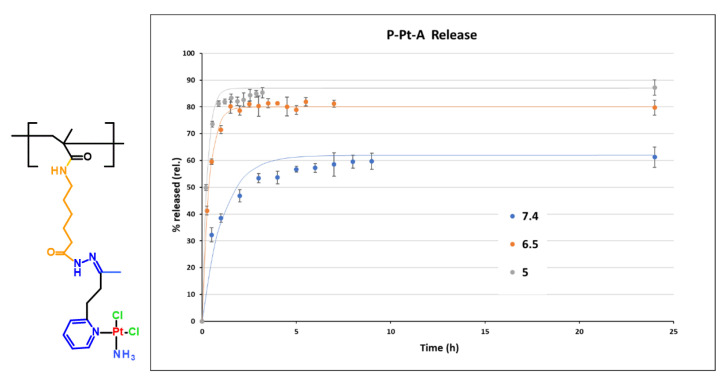
Structure of monomer unit of conjugate P-Pt-A and its hydrolytic profile at pH 7.4, 6.5 and 5. Experimental data are fitted by first-order model.

**Figure 4 pharmaceutics-17-01433-f004:**
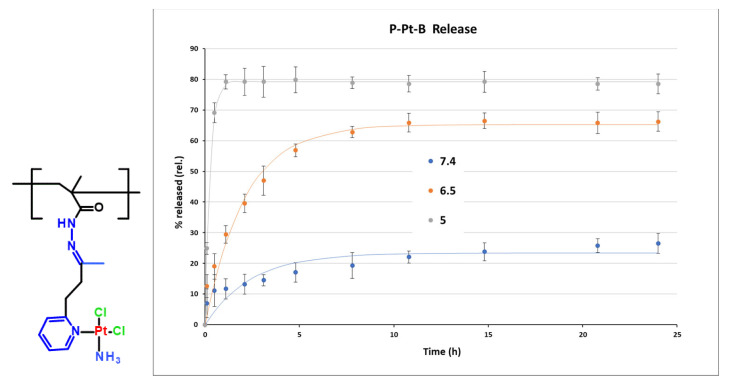
Structure of monomer unit of conjugate P-Pt-B and its hydrolytic profile at pH 7.4, 6.5 and 5. Experimental data are fitted by first-order model.

**Figure 5 pharmaceutics-17-01433-f005:**
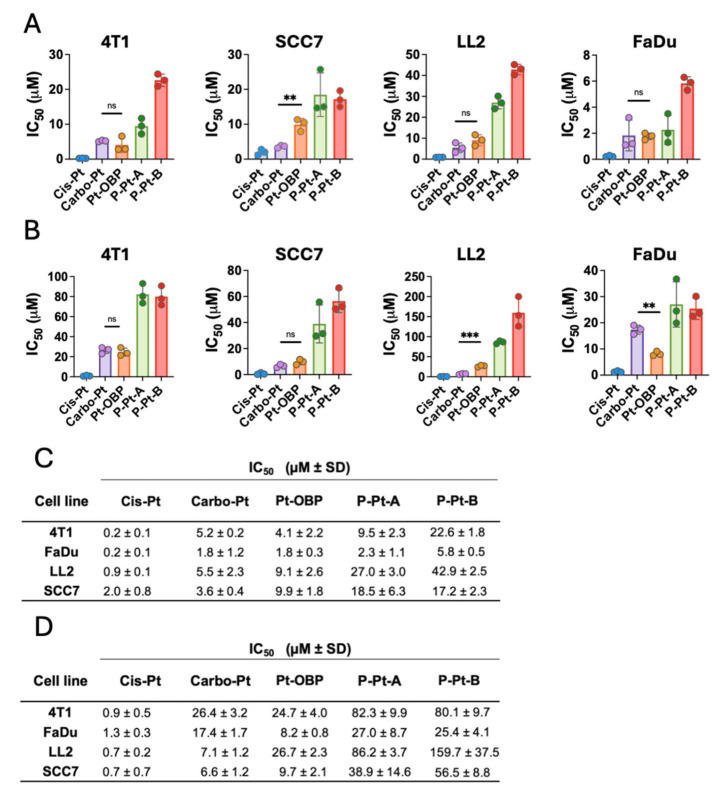
Pt-OBP shows higher or comparable cytostatic and cytotoxic activity in vitro to Carbo-Pt. Cytostatic (**A**) and cytotoxic (**B**) activities of Pt-OBP and its polymeric conjugates P-Pt-A and P-Pt-B determined using [^3^H]-thymidine incorporation and MTT assay, respectively, in 4T1, SCC7, LL2 and FaDu cell lines after 72 h of incubation. Cis-Pt and Carbo-Pt were used as reference substances. Results are shown as IC_50_ values with SD. Each experimental point represents IC_50_ value obtained in an independent experiment. Three independent experiments were performed. (**C**,**D**) Summary table of IC_50_ values ± SD. Statistical analysis was carried out using unpaired two-tailed Student’s *t*-test. Significant differences are shown *(*“*ns*” represents no significant difference; ** *p* ≤ 0.01, **** p* ≤ 0.001).

**Figure 6 pharmaceutics-17-01433-f006:**
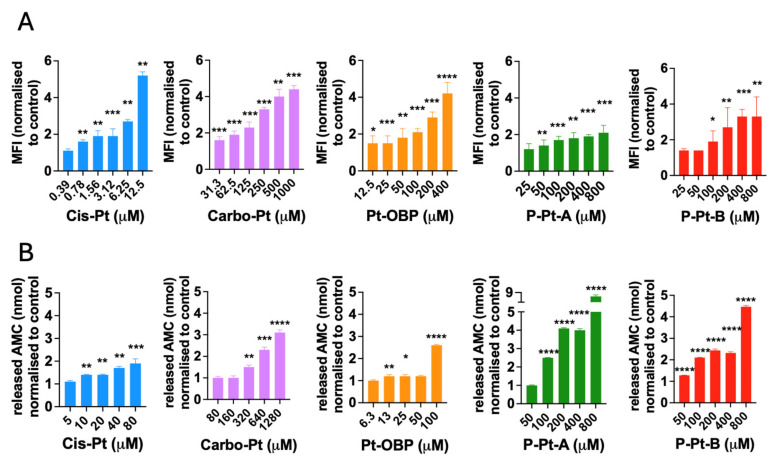
Pt-OBP and its polymeric conjugates induce DNA damage in cancer cells and activate caspase-3 in vitro. Levels of phosphorylated γ-H2AX in FaDu cells after 24 h of incubation with Pt-OBP, P-Pt-A and P-Pt-B obtained by flow cytometry analysis (**A**). Cis-Pt and Carbo-Pt were used as reference substances. Data are shown as a mean of MFI normalized to control ± SD from three independent experiments. Levels of activated capsase-3 in FaDu cells after 48 h incubation with Pt-OBP, P-Pt-A and P-Pt-B obtained by EnzChek™ Caspase-3 Activity Assay Kit with Z-DEVD-AMC substrate (**B**). Cis-Pt and Carbo-Pt were used as reference substances. Data are shown as a mean of released AMC normalized to control ± SD from three independent experiments. Statistical analysis was carried out using unpaired two-tailed Student’s *t*-test. Significant differences are shown (“*ns*” represents no significant difference; ** p* ≤ 0.05, ** *p* ≤ 0.01, **** p* ≤ 0.001, **** *p* ≤ 0.0001).

**Figure 7 pharmaceutics-17-01433-f007:**
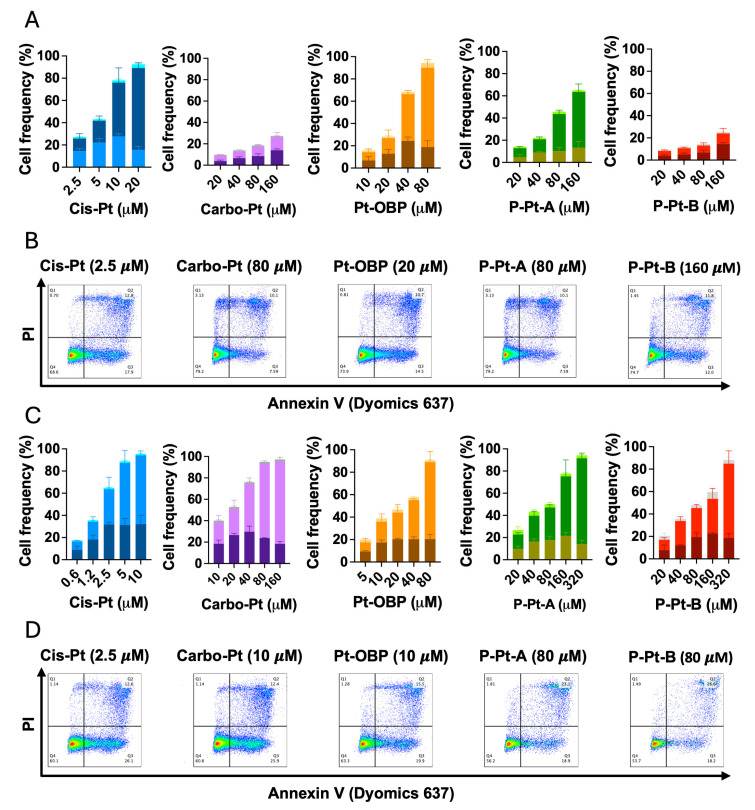
Pt-OBP and its polymeric conjugates potently induce apoptotic cell death in tumor cells in vitro. Percentage of apoptotic cells after 48 h (**A**) and 72 h (**C**) incubation with Pt-OBP, P-Pt-A and P-Pt-B obtained using Annexin V assay. Cis-Pt and Carbo-Pt were used as reference substances. Data show the distribution of the cells in different phases of the apoptotic cell death—from the bottom: PI^−^ Annexin V^+^ (early apoptosis); PI^+^ Annexin V^+^ (late apoptosis); and PI^+^, Annexin V^−^ (necrosis). Data are shown as a mean from three independent experiments ± SD. Representative dot plots are shown for each sample both after 48 h (**B**) and 72 h incubation (**D**).

## Data Availability

The original contributions presented in this study are included in the article/Appendix A. Further inquiries can be directed to the corresponding authors.

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
