# Peer review of "Stimuli-Sensitive Platinum-Based Anticancer Polymer Therapeutics: Synthesis and Evaluation In Vitro"

_pharmaceutics, 2025, doi:10.3390/pharmaceutics17111433_

Round 1
Reviewer 1 Report
Comments and Suggestions for Authors
This manuscript titled “Stimuli-sensitive Platinum-Based Anticancer Polymer Therapeutics” provides synthesis and in vitro evaluation of a novel stimuli sensitive polymer based nanotherapeutics. Two polymer–drug conjugates, P-Pt-A and P-Pt-B using hydrazone chemistry was synthesized using different spacer length and their invitro activity was evaluated in human head and neck squamous cancer cells. The study shows a sophisticated strategy, thorough methodology, and considerable technical rigor, but some minor aspects needs further clarification. Here are some specific questions:
- The conclusion suggests potential for improved pharmacokinetics and tumor accumulation via EPR, and that P-Pt-B “holds a promise for higher anticancer activity in vivo. Can authors provide particle size data in serum?
- Did authors investigate linker stability in serum containing media? I am curious how the stability of the linker is impacted in presence of different enzymes present in serum.
- The narrative states ~60% vs. ~25% at 24 h (pH 7.4) for A vs. B, but full time-course data (0–72 h) and statistical analysis are not presented. Can the authors provide cumulative release curves with SD/SE and model fitting (e.g., first-order or Higuchi) to support mechanistic differences?
- The article does not provide any safety data for this system. Please provide some toxicity data in non-cancerous or normal cell lines.
- Gating in all your apoptosis data seems to be incorrect. It might not change the conclusions heavily but please use correct gating (e.g. Figure 7).
- Please consider correcting the typo and spelling errors; section 2.9 header “Gamma-H2XA” vs. γ-H2AX; “hypopharyngeal” spelling; please ensure all units and symbols are consistent.
Reviewer 2 Report
Comments and Suggestions for Authors
Dear authors,
Thanks for your hard work and i have some basic comments and also suggested to add details in the MS as well.
1. The study you did in in vitro and what about In vivo? will provide more insight into how the drug behaves in a living organism, how it distributes through the body, and any potential side effects that might arise during long-term use. Without this, the results are limited to lab conditions, and clinical application remains uncertain.
2. What about the toxicity level of the polymer? Add more information.
3. I think and want to know, how does the polymer-drug conjugate behave in a more complex environment?
4. Could one formulation be more suited for long-term use while the other for acute treatment?
5. I suggest including a section discussing the clinical implications of your findings and the potential for advancing this technology into clinical trials, which would provide the reader with a more comprehensive understanding of the impact of your work.
6. Last how the drug concentrations were decided for the cytotoxicity assays (e.g., why those specific concentrations were chosen)?
Reviewer 3 Report
Comments and Suggestions for Authors
The paper entitled “Stimuli-sensitive Platinum-Based Anticancer Polymer Therapeutics” reports the design, synthesis, and characterization of a new cisplatin analogue, cis-[PtCl₂(NH₃)(2-(3-oxobutyl)pyridine)] (Pt-OBP), which was further covalently linked to an N-(2-hydroxypropyl)methacrylamide (HPMA) copolymer through a pH-sensitive hydrazone bond. Although the work is well designed and clearly written, addressing the following critical questions could further strengthen the manuscript.
It would be ideal to provide quantification and purity of conjugates (P-Pt-A and P-Pt-B)
- How was the platinum complex content (5.4 wt% and 4.0 wt%) determined in detail?
- Please provide (i) the full HPLC method and calibration curves used to quantify Pt-OBP on the polymer in the supporting infomation.
- Evidence that free (unbound) Pt-OBP was removed?
- Stability in biological fluids. Buffer studies are useful but insufficient. Have authors evaluated stability/release in serum or plasma (human or mouse)? If not, please provide this data or discuss how the buffer-only release experiments translate to in vivo exposure.
- Intracellular release and uptake: The paper infers intracellular release, yet no cellular uptake or intracellular Pt measurements are shown. Can you provide quantitative cellular uptake data?
- The advantages of the prodrug approach can only be validated through vivo studies, including pharmacokinetic profiling, antitumor efficacy, and toxicity comparisons between the free drug (Pt-OBP) and its polymer conjugates. Please clarify whether such investigations are currently in progress.
Overall assessment
The work addresses an important goal tumor-selective activation of platinum agents, but the current data package remains primarily chemical and in vitro. Inclusion of detailed characterization, mechanistic validation, and preliminary in vivo data or a clear plan for them would considerably strengthen the manuscript.
Round 2
Reviewer 2 Report
Comments and Suggestions for Authors
The revised version and corrections are upto mark.
Comments on the Quality of English LanguageThe english is fine
Reviewer 3 Report
Comments and Suggestions for Authors
The manuscript has been thoroughly revised, and I appreciate the authors’ detailed responses to my initial comments. I recommend the publication of this manuscript in its current form.